# In the Search of Marine Pestiviruses: First Case of Phocoena Pestivirus in a Belt Sea Harbour Porpoise

**DOI:** 10.3390/v14010161

**Published:** 2022-01-17

**Authors:** Iben Stokholm, Nicole Fischer, Christine Baechlein, Alexander Postel, Anders Galatius, Line Anker Kyhn, Charlotte Bie Thøstesen, Sara Persson, Ursula Siebert, Morten Tange Olsen, Paul Becher

**Affiliations:** 1Evolutionary Genomics Section, GLOBE, University of Copenhagen, Øster Farimagsgade 5, 1353 Copenhagen, Denmark; iben.stokholm@sund.ku.dk (I.S.); morten.olsen@sund.ku.dk (M.T.O.); 2Institute for Terrestrial and Aquatic Wildlife Research, University of Veterinary Medicine Hannover, Werftstr. 6, 25761 Büsum, Germany; ursula.siebert@tiho-hannover.de; 3Institute for Medical Microbiology, Virology and Hygiene, University Medical Center Hamburg-Eppendorf, 20251 Hamburg, Germany; nfischer@uke.de; 4Institute of Virology, University of Veterinary Medicine Hannover, Bünteweg 17, 30559 Hannover, Germany; christine.baechlein@tiho-hannover.de (C.B.); alexander.postel@tiho-hannover.de (A.P.); 5Marine Mammal Research, Department of Ecoscience, Aarhus University, Frederiksborgvej 399, 4000 Roskilde, Denmark; agj@ecos.au.dk (A.G.); lky@ecos.au.dk (L.A.K.); 6Fisheries and Maritime Museum, Tarphagevej 2, 6710 Esbjerg, Denmark; bt@fimus.dk; 7Swedish Museum of Natural History, Department of Environmental Research and Monitoring, 104 05 Stockholm, Sweden; sara.persson@nrm.se

**Keywords:** *Phocoena* pestivirus, Pinnipeds, cetaceans, marine mammals, viral phylogeny, Bayesian phylogenetic analysis, virus evolution

## Abstract

Pestiviruses are widespread pathogens causing severe acute and chronic diseases among terrestrial mammals. Recently, Phocoena pestivirus (PhoPeV) was described in harbour porpoises (*Phocoena phocoena*) of the North Sea, expanding the host range to marine mammals. While the role of the virus is unknown, intrauterine infections with the most closely related pestiviruses— Bungowannah pestivirus (BuPV) and Linda virus (LindaV)—can cause increased rates of abortions and deaths in young piglets. Such diseases could severely impact already vulnerable harbour porpoise populations. Here, we investigated the presence of PhoPeV in 77 harbour porpoises, 277 harbour seals (*Phoca vitulina*), grey seals (*Halichoerus grypus*) and ringed seals (*Pusa hispida*) collected in the Baltic Sea region between 2002 and 2019. The full genome sequence of a pestivirus was obtained from a juvenile female porpoise collected along the coast of Zealand in Denmark in 2011. The comparative Bayesian phylogenetic analyses revealed a close relationship between the new PhoPeV sequence and previously published North Sea sequences with a recent divergence from genotype 1 sequences between 2005 and 2009. Our findings provide further insight into the circulation of PhoPeV and expand the distribution from the North Sea to the Baltic Sea region with possible implications for the vulnerable Belt Sea and endangered Baltic Proper harbour porpoise populations.

## 1. Introduction

Pestiviruses comprise highly diverse and widespread RNA viruses infecting a broad range of mammalian hosts [1,2]. So far, 11 species have been approved by the International Committee on Taxonomy of Viruses (ICTV) including important pathogens such as bovine vial diarrhea viruses (BVDV-1 and BVDV-2), classical swine fever virus (CSFV) and border disease virus (BDV) [3]. These viruses can result in severe diseases in farm animals and cause significant economic losses [4,5]. During the past decade, the genus *Pestivirus* has experienced a rapid expansion of known species with the detection of novel viruses within the orders of *Artiodactyla*, *Chiroptera*, *Rodentia* and *Pholidota* [6]. A number of novel pestiviruses have been detected in pigs [7], sheep and goats [8,9,10,11,12], bats (*Scotophilus kuhli* and *Rhinolophus affini*) [13,14], rodents [14,15], pangolins (*Manis javanica* and *Manis pentadactyla*) [16] and harbour porpoises (*Phocoena phocoena*) [17], suggesting a total number of 19 pestivirus species termed *Pestivirus A* through *Pestivirus S* [6]. The recent discovery of porpoise pestivirus (Phocoena pestivirus, PhoPeV) expands the host range of pestiviruses from terrestrial to marine mammals.

Pestiviruses are enveloped positive-sense single-stranded RNA viruses with a genome length of ~12,000–16,500 nucleotides encompassing a single open reading frame (ORF). The ORF encodes 12 mature proteins (N^pro^, C, E^rns^, E1, E2, p7, NS2, NS3, NS4A, NS4B, NS5A and NS5B) and is flanked by untranslated regions (UTRs) at each end [2,6,18]. The recently described porpoise pestivirus lacks the coding region of the N-terminal autoproteinase N^pro^ found in all other known pestiviruses [17]. The protein is implicated in the immunosuppression of pestivirus-infected cells by inhibiting the activation of the interferon-1 (IFN-1)-mediated cellular innate immune response [2]. It has been speculated that the loss of this specific genomic region might be due to an adaption to the cetacean immune system [17]. However, so far, there is no knowledge about the pathogenesis of PhoPeV and its potential impact on the health of porpoise populations. In terrestrial mammals, pestiviruses can be transmitted horizontally and vertically [2]. Horizontal transmission through direct and indirect contact with secretions and excretions can result in acute disease, immunosuppression or inapparent infections. In addition, vertical transmission to the fetus through intrauterine infection during gestation can result in abortions, fetal malformations and stillbirths as well as a specific acquired immunotolerance to the infecting virus and persistent infections of the offspring [2]. Persistently infected animals are unable to produce a humoral and cellular immune response against the virus, resulting in lifelong infection of seronegative animals with continuous shedding of large amounts of virus [2,18]. Moreover, different clinical conditions have been associated with infections of pregnant cattle, sheep and pigs by the widely distributed pestiviruses BVDV, BDV, CSFV and atypical porcine pestivirus (APPV), as well as by the unique Bungowannah pestivirus (BuPV) and Linda virus (LindaV) [2,7,19,20,21]. In pregnant sows infected with such pestiviruses, reproductive failure, as well as chronic wasting, myocarditis, congenital tremor and other nervous system dysfunctions of newborn piglets, have been described. Some of these traits could possibly occur in marine mammals infected by the novel porpoise pestivirus.

The porpoise populations inhabiting the greater Baltic Sea region comprise three populations; a North Sea–Skagerrak–Kattegat population, a Belt Sea population and a Baltic Proper population [22]. These are all impacted by bycatch, prey depletion and chemical and/or noise pollution, and the Belt Sea and Baltic Sea populations are listed as vulnerable and endangered, respectively [22,23,24]. As such, the introduction of a viral disease with a range of possible health implications could be devastating. However, to date, it is not known whether porpoise pestiviruses occur in the Baltic Sea region and if so, how they relate to porpoise pestivirus strains recently detected in the North Sea. Here, we conducted the first screening for Phocoena pestivirus in harbour porpoises collected in the Baltic Sea regions of Kattegat, the Danish Belt Sea, the Arkona Basin and the Bornholm Basin. In addition, to search for novel marine mammal hosts, we also screened harbour seals (*Phoca vitulina*), grey seals (*Halichoerus grypus*) and ringed seals (*Pusa hispida*) from the Baltic Sea and North Sea regions. We report the first isolation and full genome sequence of a pestivirus strain from a harbour porpoise collected along the coastline of Zealand in Denmark in 2011. Phylogenetic investigations were conducted on the divergence between the Baltic and recently published North Sea pestivirus sequences. Additional analyses of the divergence between porpoise pestivirus and related species (represented by Bungowannah virus and Linda virus) were made to further elucidate the origin of porpoise pestivirus and the transmission of pestiviruses between land and sea.

## 2. Materials and Methods

### 2.1. Sample Collection and Virus Detection

Tissue samples (lung, spleen and reproduction organs) were collected from 77 porpoises stranded between 2007 and 2019 in the Baltic Sea and North Sea regions (Appendix A). In addition, to examining the potential occurrence of pestiviruses in sympatric marine mammal species, lung tissue from 85 harbour seals, 129 grey seals, and 63 ringed seals collected between 2002 and 2019 were included in the screenings (Appendix A). All samples were collected during necropsies and stored at −20 °C (Danish and Swedish samples) or −80 °C (German samples).

### 2.2. Virus Extraction and Screening

Extractions of viral RNA from 77 harbour porpoises and 277 seals were made on individual lung (*n* (porpoises) = 44, *n* (seals) = 173), pooled lung (*n* (porpoises) = 13, *n* (seals) = 104) and mixed organ tissue (*n* (porpoises) = 20) samples using the IndiMag Pathogen Kit w/o plastics (Cat.-No.: SP947257) with the homogenization of the tissue samples in the BeadMill 24 (Thermo Fisher) using the Lysing Matrix M (mpbio) and buffer RA1 + β-mercaptoethanol (Macherey/Nagel). Upon extraction, individual samples were combined in pools of 3–5 individuals. To establish a broadly reactive PCR, three primer pairs were designed based on the available PhoPeV sequences, as well as sequences of the related BuPV and LindaV. As no PhoPeV genome-positive material was available, the primers were evaluated using BuPV genome-positive RNA preparations in a log10 dilution series. All three primer pairs resulted in specific amplicons. To be able to investigate the samples by real-time RT-PCR, analyses were performed with the QuantiTect SYBR Green RT-PCR Kit (Qiagen). The primers marinePV_204fw (5′-GTRCYACYGGWAAGGATCACCC-3′) and marinePV_340rev (5′-CGCCGGCATCCTATCAGACTG-3′) targeting a 137 base pairs (bp) fragment located within the 5′ UTR of the genome detected the BuPV genomic RNA with the highest sensitivity and were used for PhoPeV genome screening in the organ samples. Each reaction consisted of 12.5 µL QuantiTect SYBR Green RT-PCR Mastermix, 1 µL forward primer (20 pmol) marinePV_204fw, 1 µL reverse primer (20 pmol) marinePV_340rev, 0.25 µL reverse transcriptase, 5.25 µL H_2_O and 5 µL RNA. Cycling conditions were set to 50 °C at 30 min, 95 °C at 15 min, 40 cycles of 15 s at 95 °C, 30 s at 58 °C and 30 s at 72 °C.

### 2.3. Full Genome Sequencing and Phylogenetic Analyses

The genome sequence of the PhoPeV strain 43720 detected in lung, spleen and ovary tissue samples of a harbour porpoise collected in Denmark (Zealand) in 2011 (sample ID 43720; Appendix A; accession number: OK272505) was determined by next-generation sequencing on an Illumina HiSeq as previously described [25,26]. In order to investigate the origin and phylogenetic relationship of porpoise pestiviruses, different alignments were constructed based on the novel porpoise pestivirus sequence determined in this study and data available in public repositories. The first alignment was made to investigate the phylogenetic relationship between our novel porpoise pestivirus sequence, and the sequences detected in harbour porpoises from the North Sea. The alignment consisted of eight partial PhoPeV genomes covering the 5′ UTR and the genomic region coding for C, E^rns^, E1 and E2 proteins (2502 bp) (Appendix A). The second alignment was created to further elucidate the divergence between the related PhoPev, BuPV and LindaV sequences (Appendix A). This alignment covered the conserved regions 2 and 3, corresponding to the amino acid positions 1547–2321 and 2397–2688 numbered according to the reference sequence for BVDV-1 (strain SD-1, accession number: M96751) [3]. The alignments were selected based on initial substitution and recombination analyses of different alignments covering the partial and full-length genomes of available sequences (Appendix A).

The sequences for each alignment were imported and edited in Geneious version 9.1.8 [27], where the full and partial genomes were mapped to reference sequences of BVDV-1 (M96751) and BuPV (EF100713). The sequences were edited to the same reading frame and length corresponding to the region targeted. Alignments were generated using MUSCLE [28]. All nucleotide alignments were tested for substitution saturation and recombination prior to the Bayesian phylogenetic analyses. Recombination analyses were conducted in GARD [29] using the datamonkey.org server [30] with general discrete site-to-site variation and three rate classes using the HKY85 likelihood ratio test to determine the significance of possible recombination sites (www.datamonkey.org). Substitution saturation tests were first performed using DAMBE and by evaluating plots of transitions and transversions prepared in R v.4 [31] using the packages “ape” [32] and “phylotools” [33] with the code adopted from Knudsen et al. (Appendix A) [34]. As substitution saturation was detected on the 3rd codon position, for the second alignment (Appendix A), all 3rd codon positions were removed using FASconCAT-G v1.02 [35] (2134 bp) to exclude potential bias.

The two nucleotide alignments were partitioned into 1st, 2nd and 3rd codon positions and 1st, 2nd codon positions, respectively and imported to BEAST v2.6.3 [36], where divergence time estimates and phylogenetic trees were generated using tip dates (sampling years) as time calibrations. Initial analyses with ModelTest-NG found TN93 (Tamura-Nei, 93) to be the best fit for the porpoise pestivirus alignment and TIM2 to be the best fit for the porpoise, BuPV and LindaV pestivirus alignments (Appendix A). Initial BEAST analyses were conducted on nucleotide and amino acid alignments with a strict clock with a chain length of 20,000,000 Markov chain Monte Carlo (MCMC) (Appendix A). Model selection was made based on marginal likelihood values generated through the path sampling analyses of different combinations of clock models (strict clock, lognormal relaxed clock and exponential clock) [37] and different tree models (coalescence constant, coalescence exponential growth, coalescence Bayesian skyline and birth death model) [38,39,40]. The analyses were made using a chain length of 1,000,000 generations, an alpha set to 0.3, a burn-in percentage of 50%, a pre-burn-in of 10,000 and a number of steps set to 120. In addition, the BEAST analyses of selected alignments were run with an MCMC chain length set to 200,000,000 generations, with sampling every 2000 iterations and a burn-in percentage of 10%. All log files were evaluated in Tracer v1.6.0 [41] to ensure an effective sample size (ESS) of >200 for all parameters. Finally, test runs sampling from the prior setup and excluding the alignment data were evaluated to ensure that the results were not an artifact of disagreements among the priors comprising the model.

## 3. Results and Discussion

### 3.1. PhoPeV Pestivirus Detected in the Baltic Sea Region

In this study, we screened for porpoise pestivirus in 77 harbour porpoises and 277 seals collected along the Baltic and North Sea coastlines of Denmark, Germany and Sweden between 2002 and 2019 (Appendix A). One positive case was detected among the harbour porpoises; a juvenile female (ID 43720) found stranded on the coast of Zealand, Denmark, on the 4 February 2011 (Figure 1 and Appendix A). This finding represents the first detection of PhoPeV in the Baltic Sea region and expands the distribution range of porpoise pestiviruses from the North Sea harbour porpoise population to the Danish Belt Sea population. None of the seals tested positive for a PhoPeV-related pestivirus. However, the recent expansions of the pestivirus host range from *Artiodactyla* to *Chiroptera*, *Rodentia* and *Pholidota* suggest that it is broader than previously anticipated. As such, it remains to be tested if more distantly related pestiviruses circulate among other marine mammal species.

Based on previous findings of viral RNA among animals collected in 2001, 2003–2005, 2008 and 2012 [17], it seems reasonable to assume that PhoPeV was widespread in the North Sea between 2001 and 2012 or possibly circulated as an endemic virus among harbour porpoises of the area. In our study, 27 animals including one individual from the North Sea originate from this period (Figure 1 and Appendix A). No positive animals were found among the 49 harbour porpoises collected between 2017 and 2019, but it is uncertain whether this owe to the small sample size or if the virus was not circulating in the population during this time period. All tissues analysed in this study were collected from stranded animals, and as such, this survey cannot contribute to prevalence estimations in the healthy population.

The PhoPeV-positive individual reported in this study was well nourished with a blubber thickness of 27–36 mm and a length and a weight of 117 cm and 31.2 kg, respectively. No signs of inflammation were noted during necropsy. Based on the findings of foam in the bronchi and trachea and fresh wounds on its fins and body, it was concluded that the most likely cause of death was drowning by entanglement. The detection of pestivirus RNA in lung (Ct: 20.9), spleen (Ct: 20.8) and reproductive organs (Ct: 22.1) can be the result of acute or persistent pestivirus infection. Taking into account the lack of necropsy findings and a considerably higher probability of encountering lifelong persistent pestivirus infections, a persistent infection appears more likely. As persistently infected animals stay seronegative, future surveys based on antibody prevalence should be supported by screenings for viral RNA or antigens.

### 3.2. Determination of the Genome Sequence of PhoPeV Strain 43720 and the Phylogenetic Relationship among Porpoise Pestiviruses

The genome sequence of the PhoPeV strain 43720 was determined by high-throughput sequencing using pooled RNA extracted from lung, spleen and ovary tissue samples of a harbour porpoise collected in Denmark (Zealand) in 2011 (accession number: OK272505). The obtained sequence (mean coverage: 2800) of PhoPeV strain 43720 comprises 11,872 nucleotides (nt). It contains one large ORF and 5′ and 3′ UTRs of 374 nt and 212 nt, respectively. The ORF is 11,286 nt long and encodes a polyprotein encompassing 3761 amino acids. The comparison with available complete pestivirus sequences including previously reported sequences of PhoPeV strains from the North Sea revealed that the ORF and 3′ UTR sequences of PhoPeV strain 43720 are complete, while eight highly conserved nucleotides escaped the sequencing at the 5′ terminus of the viral genome. The comparison of genome sequences revealed that PhoPeV strain 43720 is most closely related to other PhoPeV genomic sequences (>97% identity), whereas the identities with all other pestivirus sequences were below 71%. As previously described for the PhoPeV strains obtained from animals stranded at the North Sea, the newly generated sequence of PhoPeV strain 43720 lacks the pestivirus-specific N^pro^-encoding sequence [17].

Bayesian phylogenetic analyses were conducted on all currently available PhoPeV sequences to elucidate the origin of the porpoise pestivirus we detected in the Baltic Sea region. The phylogenetic analysis placed the new sequence together with genotype 1 sequences of harbour porpoises collected in the Dutch part of the Wadden Sea in 2001, 2005, 2008, 2012 and 2014. The estimated time to the most recent common ancestor (tMRCA) varied, depending on the model setup with the mean estimates ranging from 1880 to 1996 (Appendix A). However, the likelihood values provided the highest support for model 8 with an estimated tMRCA of all known PhoPeV sequences between 1986 and 2001 and a recent divergence (~2009 (2006–2011)) between our Belt Sea and previously published North Sea PhoPeV sequences obtained from animals collected between 2005 and 2009 (Figure 2 and Appendix A). Thus, our results support a rather recent exchange of PhoPeV between North Sea harbour porpoises and the harbour porpoise collected in the Baltic Sea region. The finding is significant, as the introductions or continuous circulation of PhoPeV could be yet another factor causing declines in the vulnerable Belt Sea and endangered Baltic Proper harbour porpoise populations.

### 3.3. Phylogenetic Relationship of Terrestrial and Marine Pestiviruses

Investigations of the phylogenetic relationship and divergence between porpoise pestivirus and the most closely related terrestrial BuPV and LindaV sequences were conducted to shed light on the origin of porpoise pestivirus and the dispersal of pestiviruses from terrestrial to marine hosts. Unfortunately, the detection of recombination and substitution saturation in several parts of the genome limited our final alignment to cover the 1st and 2nd codon positions of the conserved regions 2 and 3. The phylogenetic analyses using different models and clocks yielded inconsistent results with regard to the tree topology and divergence times (Appendix A; Figure 3A,B). The path sampling analyses found the highest likelihood values for the relaxed lognormal clock in combination with a coalescence exponential tree model (Appendix A). However, the topology and divergence estimates from this model suggested a close relationship between PhoPeV and LindaV and a divergence time in the late 20th century (Figure 3A; Appendix A), which is inconsistent with previous analyses of pestivirus evolution [16,17]. To check for effects of data reductions associated with the removal of the 3rd codon positions, we compared analyses including and excluding the substitution-saturated 3rd codon positions separately but found no significant difference in the divergence estimates and topology. Moreover, test runs of the relaxed clock analyses only sampling from the prior provided similar time estimates as runs with data, with no apparent discrepancies in the model setup that could artificially drive these early estimates. However, the data set is limited to just four PhoPeV, three BuPV and two LindaV sequences which taken together with the alignments’ limited coverage of variable regions may contain insufficient information for the relaxed clock models to be informative. In contrast to the results of the relaxed clock model, the strict clock model supported a close relationship between BuPV and LindaV with porpoise pestivirus as an outgroup to these (Figure 3B), corresponding to previously reported phylogenetic relationships between the viruses [16,17]. The divergence time estimates provided by the strict clock model varied, depending on the input data and the tree model, but most pointed to a divergence of PhoPeV from other pestiviruses between the 17th and the 20th centuries (Figure 3; Appendix A). However, based on the path sampling analysis providing the highest support for a relaxed clock and the discussed implications of the limited data set, we regarded these results as inconclusive.

PhoPeV, BuPV and LindaV are each other’s closest relatives among known pestiviruses but only share 60% homology at the amino acid level [17]. Thus, future studies on the origin of marine pestiviruses would benefit from the detection and analysis of more closely related strains, i.e., “missing links”, which would allow for the creation of data sets holding less recombination and substitution saturation and aid the generation of more accurate time estimates. Moreover, recent analyses of the divergence between related morbilliviruses, Measles virus and Rinderpest virus, utilized codon-based models and ancient virus strains to obtain more stable and reliable results [42,43]. While the detection of ancient wildlife pestiviruses currently seems unlikely, future investigations into the evolutionary history of these viruses would benefit from an extended data set including additional genome sequences representing a larger number of diverse porpoise, Bungowannah and Linda pestiviruses. Such additional analyses could be carried out on full-genome codon-based models (excluding recombinant sites) [43]. As seen with other studies on virus evolution, it is likely that the addition of additional, novel and historical strains will similarly push the estimated divergence between terrestrial and marine pestiviruses back in time [43,44].

### 3.4. Pestiviruses in the Marine Ecosystem

The detection of PhoPeV in a harbour porpoise from Zealand, Denmark, expands the distribution range from the North Sea to the Belt Sea populations. Its presence gives cause for concern, as infections might negatively impact the reproductivity of the vulnerable Belt Sea and endangered Baltic Proper harbour porpoises. The Belt Sea case could be a result of a single introduction of a widespread virus or the observation of an endemically circulating virus detected in a single individual. Additional findings of virus, viral genomes, antigens or antibodies are needed to further evaluate the circulation and presence of PhoPeV among harbour porpoise populations in this region and elsewhere to determine the impact of the virus. Therefore, we suggest that harbour porpoise health-monitoring programs should include this putative pathogen.

Too little is currently known about PhoPeV to establish the origin and dispersal of the virus. While only recently detected in porpoises, pestiviruses might have been circulating in the marine ecosystem for centuries. The absence of the N^pro^ protein-coding region in the PhoPeV genome clearly distinguishes it from other pestiviruses and suggests that there might be yet unidentified “missing links” among PhoPeV and related pestiviruses. This assumption is further supported by the disparate distribution of BuPV and LindaV, which represent unique viruses and have only been found in pig herds on geographically distant farms in Australia and Austria, respectively [7,45], as well as by the recent discovery of the related Dongyang pangolin virus (DYPV) in China [16]. It has been speculated that human-mediated dispersal by livestock transport or a contaminated vaccine could have promoted virus spread across large distances [46]. However, as BuPV and LindaV have not been detected in other domestic pig herds or in wild boar populations, it seems more likely that the emergence of these two porcine viruses were caused by spillover from unknown (wild) reservoirs [47,48,49]. A more complete understanding of the origin, evolution and spread of PhoPeV, BuPV and LindaV awaits the discovery of related pestiviruses in marine and terrestrial mammals.

### 3.5. Isolation of PhoPeV 43720 and Future Studies

It has been reported that PhoPeV from the North Sea could be isolated on porcine and bovine cell cultures [17]. While initial attempts to isolate the Baltic PhoPeV strain 43720 failed, virus isolation was finally successful by the inoculation of porcine SPEV kidney cells with dilutions of organ suspensions. The availability of PhoPeV isolates will facilitate the establishment of assays for the detection of PhoPeV-specific antibodies and allow the determination of prevalence and cross-reactivity with related pestiviruses. Future studies will focus on the characterization of biological properties of PhoPeV and include investigations on the host range, viral entry and replication in tissue culture cells to enhance our knowledge about the biology of this first marine pestivirus.

## Figures and Tables

**Figure 1 viruses-14-00161-f001:**
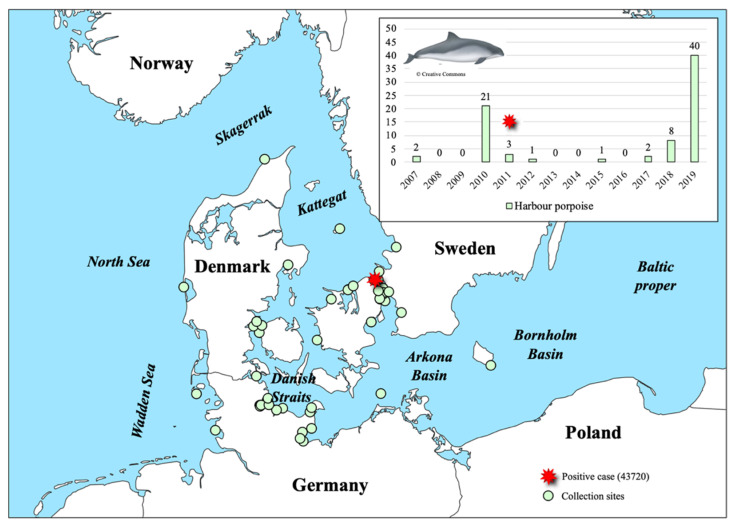
Map and histogram illustrating collection sites (green dots) and distribution of 77 harbour porpoises screened per year between 2007 and 2019. The red star indicates the collection site of the Phocoena pestivirus (PhoPeV)-positive harbour porpoise (ID: 43720 and collection date: 4 February 2011).

**Figure 2 viruses-14-00161-f002:**
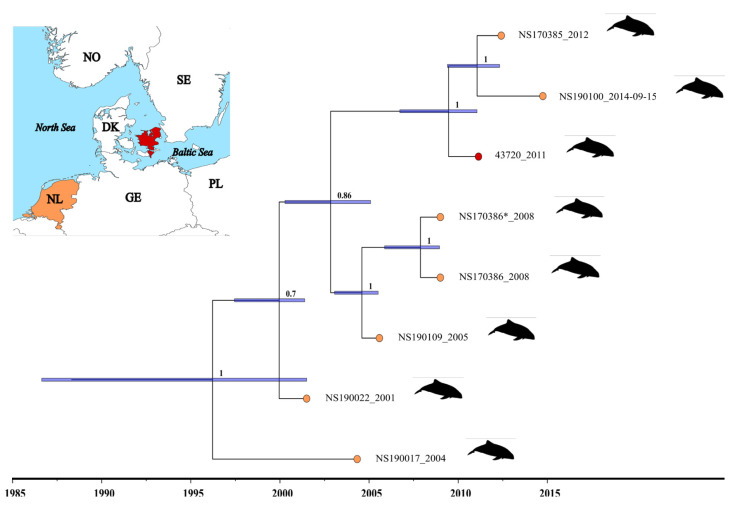
Time-calibrated Bayesian phylogenetic tree of the Baltic PhoPeV sequence (43,720) described here and seven recently published PhoPeV sequences from the Netherlands collected between 2001 and 2014 (GenBank accession number: MK910230-37). The tree was generated in BEAST 2.5.2 based on the concatenated nucleotide alignment comprising the 5′ untranslated regions (UTR) and the regions coding for C, E^rns^, E1 and E2 proteins. The tree was generated using a relaxed exponential clock and a coalescence constant tree prior (Appendix A). The colour-coded circles at the tips indicate the countries (the red colour represents Denmark, and the orange colour represents the Netherlands) where the animals were collected. The node bars represent the highest posterior density (HPD) interval of the time estimates, while the tree support is shown as posterior values at each node.

**Figure 3 viruses-14-00161-f003:**
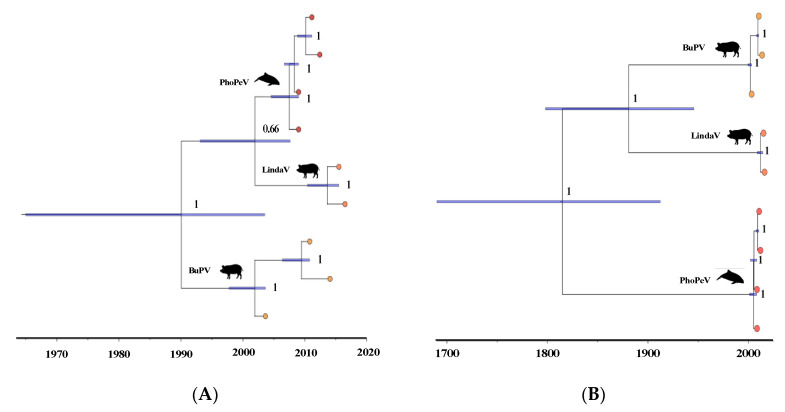
Time-calibrated Bayesian phylogenetic trees of available PhoPeV, Bungowannah pestivirus (BuPV) and Linda virus (LindaV) sequences (Appendix A) covering conserved regions 2 and 3 corresponding to the amino acid positions 1547–2321 and 2397–2688 numbered according to the reference sequence for BVDV-1 (strain SD-1) (accession number: M96751) [3]. The trees were generated in BEAST 2.5.2 under two different model setups. (**A**) Tree generated using a relaxed clock lognormal and an exponential coalescence tree prior. This tree was the most strongly supported tree based on the path sampling analysis (Appendix A). (**B**) Tree generated using a strict clock and a Bayesian skyline tree prior. This model set up showed the strongest support among models using a strict clock (Appendix A). The node bars represent the highest posterior density (HPD) interval of the time estimates, while the tree support is shown as posterior values at each node.

## Data Availability

The data sets are submitted together with the study. The novel PhoPeV sequence has been submitted to GenBank (accession number: OK272505).

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
