# Peer review of "In the Search of Marine Pestiviruses: First Case of Phocoena Pestivirus in a Belt Sea Harbour Porpoise"

_viruses, 2022, doi:10.3390/v14010161_

Round 1

Reviewer 1 Report

The study is interesting, especially as the collection of samples is unique. It has a significant scientific value for the marine virology. Perhaps only lacks some insites into the pathology of pestivirus infections. What is the meaning of those infections for the species health, nature conservation etc.? Perhaps this should be discussed in the text.

Author Response

Thank you for your feedback. Your question is spot on and ongoing experiments aim to elucidate this question further. Yet, at the moment we can only speculate that the virus may inflict some of the same pathogenesis found among its closest known relatives; the LindaV and BuPV This is stated e.g. in line 88.

Reviewer 2 Report

Review manuscript ID: viruses-1543523

In the present manuscript, Stokholm and his colleagues screened 77 harbor porpoises, 277 grey seals, harbor seals, and ringed seals for pestiviruses, looking primarily for the porpoise pestivirus sequence. One positive case was discovered, providing further evidence of harbor porpoise pestivirus from a harbor porpoise from the Baltic Sea. There was not a single detection from any seal species, which made up a large proportion of the samples. Of course, it is fine to check if other hosts are infected by the harbor porpoise pestivirus or if other related viruses circulate in the pinnipeds, but linking these negative data to the positive detection in the harbor porpoise does seem somewhat indiscriminate. The same applies to the title, which is misleading with respect to the data presented. It is not about pestivirus in marine mammals in the Baltic and North Seas in general, but about another detection of harbor porpoise pestivirus in harbor porpoises and the lack of detection in seals. Could the new virus strain not be isolated? Then this should also be communicated. But if the virus was isolated, at least a preliminary biological characterization should also be connected, which should include culture cells, virus growth and antigenicity, in order to advance further research on the virus. Overall, the manuscript contains data worth sharing.

Main points:

  1. The discussion in lines 191-201 is not useful, since the sample size does not really allow any conclusions or comparisons. It would be desirable to calculate the necessary sample size to corroborate the existing weak data on prevalence. One point that definitely needs to be included before publishing the data is the bias in the data, which were generated exclusively from stranded dead animals and thus do not allow any conclusions about prevalence in the healthy population.
  2. Line 226: “…while the sequence of strain 43720 lacks eight highly conserved nucleotides at the 5’ terminus of the viral genome.” This should be better explained. Are the nucleotides missing from the isolate, could they not be read during sequencing? Wouldn't it be important to verify the nucleotides when talking about a complete genomic sequence?
  3. The entire chapter "Phylogenetic relationship of terrestrial and marine pestiviruses" including the illustrations seems very speculative and methodologically rather questionable. Besides the small amount of data available for the calculations, the calculations are mainly dependent on the assumed mutation rate. However, this was not determined for any of the viruses in their hosts. This should be communicated much more clearly.
  4. As I read the first manuscript on a porpoise pestivirus for this review, it is also noticeable that an important point was completely omitted in this manuscript. What about virus isolation? Does the virus behave similarly to related strains? This point of characterization would be much more obvious than looking for the virus in seals, wouldn't it?

Minor points:

  1. Please re-phrase line 67, 68. The acquired immunotolerance of the fetus is certainly not causing abortion. Please cite relevant references for the outcome of vertical transmitted pestiviral disease.

Author Response

Referee: 2

In the present manuscript, Stokholm and his colleagues screened 77 harbor porpoises, 277 grey seals, harbor seals, and ringed seals for pestiviruses, looking primarily for the porpoise pestivirus sequence. One positive case was discovered, providing further evidence of harbor porpoise pestivirus from a harbor porpoise from the Baltic Sea. There was not a single detection from any seal species, which made up a large proportion of the samples. Of course, it is fine to check if other hosts are infected by the harbor porpoise pestivirus or if other related viruses circulate in the pinnipeds, but linking these negative data to the positive detection in the harbor porpoise does seem somewhat indiscriminate.

>> Thank you for your valuable feedback. In this manuscript we do not seek to link the negative cases in the seals to the positive case detected among the harbour porpoises. We mention the seals as they were a part of the initial screening and we with this unique data set found it relevant to state that we gave it a try. We reformulated the sentence at line 243-250 to further separate the results.

The same applies to the title, which is misleading with respect to the data presented. It is not about pestivirus in marine mammals in the Baltic and North Seas in general, but about another detection of harbor porpoise pestivirus in harbor porpoises and the lack of detection in seals.

>> We have corrected the title to “In the search for marine pestiviruses: First case of phocoena pestivirus in a Belt Sea harbour porpoise”

Could the new virus strain not be isolated? Then this should also be communicated. But if the virus was isolated, at least a preliminary biological characterization should also be connected, which should include culture cells, virus growth and antigenicity, in order to advance further research on the virus. Overall, the manuscript contains data worth sharing.

>> Thank you for this comment. Indeed, while preparing the submitted manuscript, we were able to isolate the virus. However, a biological characterization (as suggested by this reviewer) was not in the scope of the present manuscript and the results will be part of a future comprehensive study including host cell range, virus growth, antigenicity and other selected biological properties.

Main points:

  1. The discussion in lines 191-201 is not useful, since the sample size does not really allow any conclusions or comparisons. It would be desirable to calculate the necessary sample size to corroborate the existing weak data on prevalence. One point that definitely needs to be included before publishing the data is the bias in the data, which were generated exclusively from stranded dead animals and thus do not allow any conclusions about prevalence in the healthy population.

>> This comment was very helpful. It was not our intention to make conclusion about prevalence in the population. To clarify this point we have added the following sentence to the end of this paragraph: ”All tissues analysed in this study were collected from stranded animals and as such this survey cannot contribute to prevalence estimations in the healthy population.” (see lines  260 to 262 of the revised manuscript)

  1. Line 226: “…while the sequence of strain 43720 lacks eight highly conserved nucleotides at the 5’ terminus of the viral genome.” This should be better explained. Are the nucleotides missing from the isolate, could they not be read during sequencing? Wouldn't it be important to verify the nucleotides when talking about a complete genomic sequence?

>> We have corrected this sentence to make it clearer that the 8 nucleotides were not read during sequencing. As it is only a matter of 8 nucleotides, and they are considered highly conserved we do find this novel sequence is well-covered. Line 327-331.

  1. The entire chapter "Phylogenetic relationship of terrestrial and marine pestiviruses" including the illustrations seems very speculative and methodologically rather questionable. Besides the small amount of data available for the calculations, the calculations are mainly dependent on the assumed mutation rate. However, this was not determined for any of the viruses in their hosts. This should be communicated much more clearly.

>> These time calibrated phylogenetic analyses were generated using the computer program called BEAST which allow the integration of sample dates to the sequence data under a Bayesian framework. Here, the calculation of mutation rates among the viruses is an integrated part of the analysis. However, the analysis in itself is much more complex and it does not just depend on the mutation rate. The prior knowledge about the viruses is integrated through a combination on different evolutionary models. We test how the viruses have evolved by comparing different scenarios of evolution to the actual data (e.g. including different molecular clocks that allow for a fixed or relaxed mutation rate between viruses and among different strains). As stated in the discussion we find that the amount of information in the current collection of LindaV, BuPV and PhoPeV is insufficient for such complex analyses, but we believe that this information provides an important point i.e. that more data is needed and care should be taken when calculating and interpreting divergence times estimates on viral phylogenies with limited data access. Furthermore, the accumulation of substitution saturation is often neglected and should be accounted for as this natural process could bias the rate by which the evolution occurs. As such it is also important to consider whether or not the phylogenetic relationship between specific viruses can be evaluated in this kind of analyses or if they are too distantly related.

We have made some corrections in the section to make our points more clear but we think that including a section on how Bayesian phylogenetic analyses work will be too much.

As I read the first manuscript on a porpoise pestivirus for this review, it is also noticeable that an important point was completely omitted in this manuscript. What about virus isolation? Does the virus behave similarly to related strains? This point of characterization would be much more obvious than looking for the virus in seals, wouldn't it?

>> Please see our third comment (page 1 in this document).

*The line numbers refer to the numbers in the tracked_changes document.

Reviewer 3 Report

The manuscript by Stokholm et al. presents a targeted investigation of the presence of porpoise pestiviruses in tissue samples from marine mammals including porpoises, harbour seals, gray seals and ringed seals, collected along the Baltic and North Sea coastlines between 2002 and 2019. Among the screened samples, the lung, slpeen and ovary tissues of a single porpoise tested positive. Combined full genome sequencing and phylogenetic analyses of the identified Phocoena pestivirus (PhoPeV) presented in the manuscript indicate that (i) it is closely related to other previously identified PhoPeV, (ii) it belongs to porpoise pestivirus genotype 1 and (iii) it has recently diverged from PhoPeV collected in the North Sea. In addition, investigation of the phylogenetic relationship with members of the pestivirus genus suggests that PhoPeV diverged from the most related Bungowannah virus (BuPV) and LINDA virus between 1700 and 1900 according to a strict clock model and that they represent a distinct branch in the genus. Finally, pathogenicity, spread and origin of the virus are discussed in the manuscript.

Overall, the manuscript provides new information regarding the distribution of marine pestiviruses and further supports previous findings revealing unique features of their genomes, i.e. absence of Npro coding sequence. The main weakness is the limited information of phylogenetic analyses that is precluded by the small number of sequences available for marine pestiviruses and the related BuPv and LindaV. 

Author Response

The manuscript by Stokholm et al. presents a targeted investigation of the presence of porpoise pestiviruses in tissue samples from marine mammals including porpoises, harbour seals, gray seals and ringed seals, collected along the Baltic and North Sea coastlines between 2002 and 2019. Among the screened samples, the lung, slpeen and ovary tissues of a single porpoise tested positive. Combined full genome sequencing and phylogenetic analyses of the identified Phocoena pestivirus (PhoPeV) presented in the manuscript indicate that (i) it is closely related to other previously identified PhoPeV, (ii) it belongs to porpoise pestivirus genotype 1 and (iii) it has recently diverged from PhoPeV collected in the North Sea. In addition, investigation of the phylogenetic relationship with members of the pestivirus genus suggests that PhoPeV diverged from the most related Bungowannah virus (BuPV) and LINDA virus between 1700 and 1900 according to a strict clock model and that they represent a distinct branch in the genus. Finally, pathogenicity, spread and origin of the virus are discussed in the manuscript.

Overall, the manuscript provides new information regarding the distribution of marine pestiviruses and further supports previous findings revealing unique features of their genomes, i.e. absence of Npro coding sequence. The main weakness is the limited information of phylogenetic analyses that is precluded by the small number of sequences available for marine pestiviruses and the related BuPv and LindaV. 

>> Thank you very much for your valuable input. We wish we had a more informative data set for the PhoPeV, BuPV and LindaV analyses but we were unfortunately limited by the sample size and the fact that a large fraction of the alignments we tested were affected by recombination and substitution saturation, mainly because the viruses are relatively distinct. Thus, future analyses would benefit from the detection of “missing links” between the viruses and calculations on divergence times between novel viruses more closely related to PhoPeV. For now, we cannot trust the time estimates and we suspect the divergence to be much older in reality, but the setup is legit, and we believe that these points are still important to discuss to further highlight the fact that care should be taken during setup and when interpreting this kind of analyses. 
